# Vector competence for Oropouche virus: A systematic review of pre-2024 experiments

Emily N. Gallichotte[1]*, Gregory D. Ebel[1], Colin J. Carlson[2]*

**1** Department of Microbiology, Immunology and Pathology, Colorado State University, Fort Collins, Colorado, United States of America, **2** Department of Epidemiology of Microbial Diseases, Yale School of Public Health, New Haven, Connecticut, United States of America

* emily.gallichotte@colostate.edu (ENG), colin.carlson@yale.edu (CJC)

## Abstract

The 2023–24 epidemic of Oropouche fever in the Americas and the associated ongoing outbreak in Cuba suggests a potential state shift in the epidemiology of the disease, raising questions about which vectors are driving transmission. In this study, we conduct a systematic review of vector competence experiments with Oropouche virus (OROV, *Orthobunyavirus*) that were published prior to the 2023–24 epidemic season. Only seven studies were published by September 2024, highlighting the chronic neglect that Oropouche virus (like many other orthobunyaviruses) has been subjected to since its discovery in 1954. Two species of midge (*Culicoides paraensis* and *C. sonorensis*) consistently demonstrate a high competence to transmit OROV (~30%), while mosquitoes (including both *Aedes* and *Culex* spp.) exhibited an infection rate consistently below ~20%, and showed limited OROV transmission. Further research is needed to establish which vectors are involved in the ongoing outbreak in Cuba, and whether local vectors and wildlife communities create any risk of establishment in non-endemic regions.

## Author summary

Oropouche virus has recently become an urgent threat to public health in Central and South America. OROV is mainly transmitted by biting midges; however, some public health agencies and scientific sources note that some mosquito species transmit the virus. We conducted a systematic review of literature prior to the current epidemic, and identified seven studies that experimentally tested the ability of vectors to become infected with, and transmit OROV (i.e., that assessed their vector competence). These studies have consistently found that biting midges become infected at higher rates than mosquitoes, which rarely transmit the virus. It is unclear which vectors are responsible for transmitting OROV in the current outbreak. Existing published data support the observation that biting midges are likely to be significant vectors compared to mosquitoes, which are comparatively incompetent. However, increased vector surveillance and pathogen testing, and additional vector competence experiments using current OROV strains, are urgently needed.

**Data availability statement:** All data are publicly available on Figshare (10.6084/M9.FIGSHARE.27157029.V1; https://figshare.com/articles/dataset/Oropouche_vector_competence_data/27157029?file=49855260)

**Funding:** This work was supported by National Science Foundation DBI 2021909, 2213854, and 2515340 (CJC). The funders had no role in study design, data collection and analysis, decision to publish, or preparation of the manuscript.

**Competing interests:** The authors have declared that no competing interests exist.

## Introduction

Oropouche virus (OROV) is a Simbu serogroup orthobunyavirus endemic to South America and parts of the Caribbean. Historically, OROV has been neglected compared to other arboviral diseases, such as yellow fever, dengue, chikungunya, or Zika. However, an ongoing epidemic of Oropouche fever has brought new attention to the virus. In the first seven months of 2024, there were more than 8,000 cases across the Americas, mostly concentrated in Brazil; at the time of writing, a second epidemic wave is ongoing in Cuba, with over 11,000 suspected cases. The scale of this outbreak may be connected to evolutionary changes in the pathogen: recent genetic analyses revealed that the OROV lineage currently circulating in Brazil is a novel reassortant containing M segment from viruses detected in the eastern Amazon from 2009-2018, and L and S segments from viruses detected in Peru from 2008-2021 [1]. *In vitro* characterization of the novel reassortant virus recently demonstrated that it replicates to higher levels than the prototypic strain in mammalian cells, and is less sensitive to neutralization by human OROV immune sera collected prior to 2016 [2]. This reassortment event may also result in changes in the clinical presentation of Oropouche fever. Prior to 2024, symptoms were generally considered similar to other febrile illnesses, and no deaths had been reported [3]; in the 2024 outbreak, however, two deaths caused by OROV were reported in healthy young women, and there have been multiple reports of miscarriage, fetal deaths, and microcephaly associated with OROV infection [4].

Surprisingly little is known about the vectors involved in the current epidemic. Current evidence suggests that, unlike many other arthropod-borne orthobunyaviruses, OROV is primarily transmitted by culicoid midges (Ceratopogoidae: *Culicoides*) rather than mosquitoes or ticks [5]. *C. paraensis* are considered the principal vector of epidemic urban OROV transmission due to their high abundance in locations of previous OROV outbreaks, and OROV isolation from *C. paraensis* multiple times during a 1975 outbreak in Brazil [6–8]. *Culicoides* are efficient vectors for many arboviruses, including other Simbu serogroup orthobunyaviruses in South America [9]. They feed on a variety of vertebrates, and three-toed sloths, birds, and non-human primates (capuchin and howler-monkeys) are thought to be the primary hosts of sylvatic OROV [10].

When OROV was first detected in 1955 in Trinidad & Tobago, over 700 mosquitoes (*Aedes*, *Wyeomyias*, *Psorophora*, *Mansonia*, *Culex*, *Anopheles*, *Haemogogous*, and other unidentified sabethines) were collected from the same area as the infected patient, but OROV was only detected in *Coquillettidia venezuelensis* (referred to as *Mansonia venezuelensis* in the paper) [11]. In 1961, OROV was isolated from a pool of *Aedes serratus* in Brazil [12]. Because *Cq. venezuelensis* and *Ae. serratus* are hematophagous mosquitoes that inhabit sylvatic environments, they were suggested as potential sylvatic vectors. *Cx. quinquefasciatus* have also been proposed as a secondary, urban, anthropophilic vector because OROV has been isolated from them multiple times [7,13]. However, in all instances of virus isolation from mosquitoes, the resulting detection rates were very low, suggesting poor susceptibility of the vector to infection. Overall, these suggest that many blood-feeding arthropods

are exposed to the virus in nature, but mosquitoes may not be meaningfully involved in transmission. The scale of the current epidemic, particularly in Cuba, has prompted speculation about a potential shift to mosquito vectors, especially since no *Culicoides* species have been reported there, but so far no observational data supports this idea [14,15].

To define the extent and outcomes of previously published vector competence experiments using OROV, we developed a standardized dataset of all pre-2024 records of vector competence experiments that studied OROV, following a previously-developed data standard [16]. The data standard contains fields for vector, virus, exposure, experimental and infection conditions, and experimental outcomes, allowing us to standardize variables, making it easier to compare across studies. Despite the large number of outbreaks over the last 50 years, there is significant uncertainty regarding which vectors are responsible for OROV transmission, including in the current epidemic. Because the virus is spreading to locations where it has not previously been detected (e.g., Cuba [17]), and could continue to spread to new locations (e.g., the United States), it is critical to understand transmission risk posed by a wide range of potential vectors. Unfortunately, we found that before 2024 there was limited experimental research testing the ability of different vectors to become infected with, and transmit OROV. The small number of studies that have been conducted demonstrate that *Culex* spp. can infrequently be infected with OROV, but transmit virus at low rates compared to *Culicoides* midges. It thus seems unlikely that mosquitoes have been a major vector of OROV thus far.

## Materials and methods

### Systematic search

A systematic search was conducted on September 17, 2024 on PubMed using the search term "Oropouche virus" (no other databases were used, nor were additional spellings or abbreviations included as search terms). No filters or limits were used. The search returned 168 publications, which were imported into Rayyan for manual screening [18]. Exclusion criteria included: reviews, news articles, commentaries, surveillance studies, experimental studies in systems other than vectors (e.g., cells, mice, etc.), etc. The following inclusion criteria were required: full text available in English; experimental OROV infections in vectors (mosquitoes or midges), and raw data must be available (e.g., number of individual vectors positive and total number tested, not derived rates) (no papers were identified that did not have raw data available). A single reviewer screened all publications. Two additional publications were identified from citation searching. One publication was frequently cited in multiple reviews stating susceptibility of multiple vectors to OROV infection [11]. This paper was identified in our original search, but had been excluded because there was no mention of experimental vector competence in the title or abstract. A second publication did not show up in our original search, but was also identified due to frequent reference in other publications [19]. There were seven publications that met all criteria and were used in our analyses.

### Data collection

Information was extracted from publications into a standardized template, following a previously-published minimum data and metadata standard [16]. Data sections include information on vectors, viruses, experimental conditions, infection conditions, and infection outcomes (specifically sample type tested, assay used to detect infection, number tested, and number positive). Information not provided in publications were left blank, and no assumptions were made about any missing or unclear information. Experiments evaluating the ability of OROV to be mechanically transmitted by vectors were not included. Authors of de Mendonça *et al.* [20] were contacted for the vector origin year which was provided. Risk of bias assessment, effect measures, reporting bias assessment and certainty assessment were not determined.

### Data analysis and statistics

All data were analyzed in GraphPad Prism Version 10.2.3.

## Results

### Systematic search and selection process

We performed a systematic review using the Preferred Reporting Items for Systematic reviews and Meta-Analyses (PRISMA) reporting guidelines. A total of 168 publications were retrieved from PubMed (see Methods for search criteria) and screened for eligibility. Abstracts were reviewed, and only five met all eligibility criteria [20–24]. Two additional publications were identified from citation searching, which met all eligibility criteria, and were included in the final sample (Fig 1) [11,19]. Experimental data were extracted from published studies into a previously described template allowing for comparisons and analyses [16].

### Summary of included studies

The seven eligible studies were published in 1961, 1981, 1982, 1987, 1991, and 2021 (n = 2 studies in 2021) (Fig 2a). Shortly after the first OROV outbreak in Trinidad in 1954, the first experimental vector competence study was published. OROV subsequently caused outbreaks in many countries in Central and South America, with many in Brazil and Peru (Fig 2a). Minimal experimental details were available and uncommon methods were used in Anderson et al. [11], making

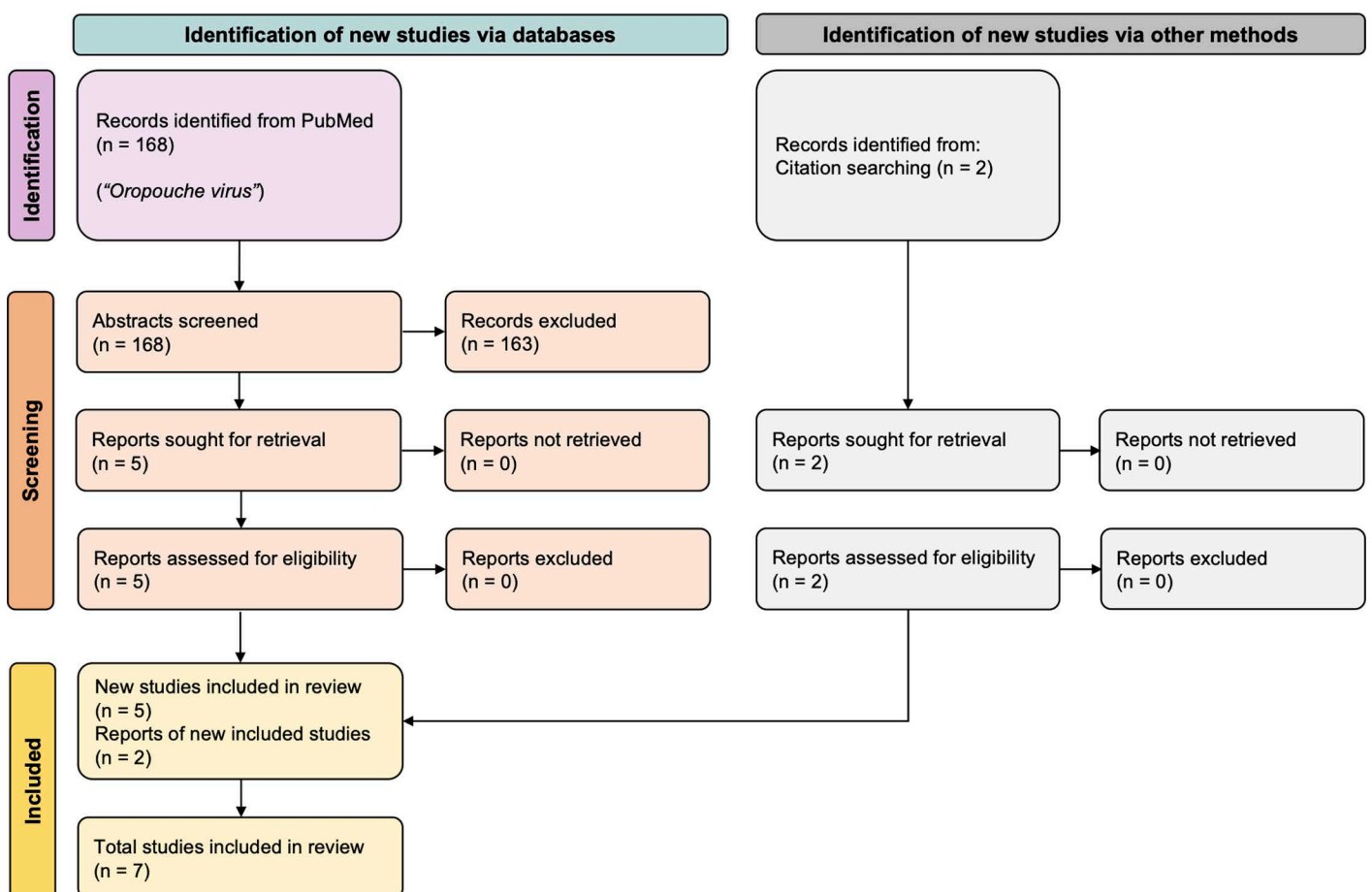

**Fig 1. PRISMA flow diagram of search according to PRISMA 2020 strategy [36].**

comparisons across studies challenging, so results from this study are discussed separately later. The six remaining studies included multiple *Aedes*, *Culex*, and *Culicoides* species, with multiple studies comparing different vectors (Fig 2b).

## Vectors used in experimental studies

Vectors used in studies were originally collected throughout the Americas and in Thailand, and in some studies, multiple species were collected from the same location (Fig 3a and 3b). Some vectors were collected from the field within a few years of study publication, however other colonies (e.g., *Cx. tarsalis* from McGregor *et al.* [23]), were colonized 60 + years prior to experimental infections and publications (Fig 3c).

A total of 2,408 individual vectors were tested in vector competence experiments; 28% *Aedes* spp., 55% *Culex* spp. and 16% *Culicoides* spp. (Fig 4a). Vectors were exposed to OROV via feeding on a live animal, artificial bloodmeal, and intrathoracic injection (Fig 4a). Of those fed on live animals, the majority were fed on infected hamsters, with a smaller proportion fed on AG129 mice, and humans (Fig 4b). Vectors were exposed to a large range of OROV infectious titers, quantified using multiple techniques, ranging from 5.2 to 9.9 $\log_{10}$ SMLD$_{50}$/mL (Fig 4c).

## Vector competence of mosquitoes and midges infected with OROV

Infection rates were calculated for all studies, with time points, virus quantification method, vector sample, and other variables combined. Of vectors infected via a live animal or artificial bloodmeal, only *Culicoides* (both *C. paraensis* and *C. sonorensis*) had infection rates >20% (Fig 5a). This was dose-dependent, as no *C. paraensis* exposed to OROV ≤5.2 $\log_{10}$ SMLD$_{50}$/mL were infected. Conversely, while *Aedes* and *Culex* spp. injected with OROV were efficiently infected, those exposed via live animal and artificial bloodmeal had infection rates <20% regardless of virus titer (Fig 5a). Despite a range of infection rates, when comparing individual experiments, *Culicoides* spp. had higher average infection rates than all mosquito species tested (Fig 5b). Similar trends are seen with dissemination rates, *C. sonorensis* have higher rates of dissemination compared to both *Aedes* and *Culex* spp. (Fig 5c).

Across studies, transmission was evaluated two ways: the detection of OROV in expectorated vector saliva, and detection of infection (virus or seroconversion) in naïve animals fed on by infected vectors (Fig 5d). *C. sonorensis* had significantly higher rates of OROV in saliva compared to *Cx. tarsalis* and *Cx. quinquefasciatus* (Fig 5d). Similarly, *C. paraensis* had higher rates of OROV transmission to naïve animals compared to *Ae. albopictus* and *Cx. quinquefasciatus*, however the confidence intervals are much larger due to the lower number of animals tested as compared to vector saliva (Fig 5d).

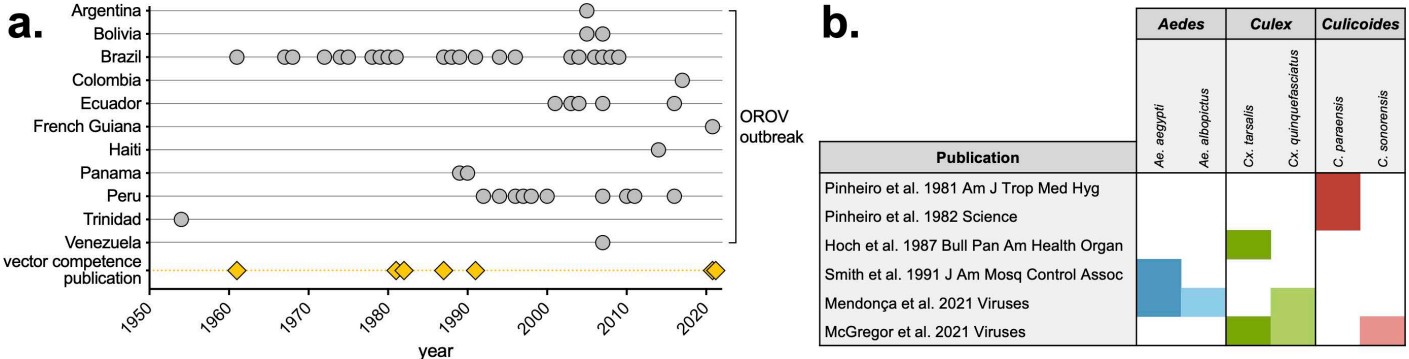

**Fig 2. OROV outbreaks and experimental vector competence publications.** a) Timeline of OROV outbreaks in countries in the Caribbean and Central and South America, modified from Tilston-Lunel, 2024 [37]. Experimental vector competence publications from PRISMA review are shown by publication date. b) Vector species (*Aedes, Culex* and *Culicoides* spp.) tested for each of the six papers identified through the PRISMA search (Fig 1). [19–24].

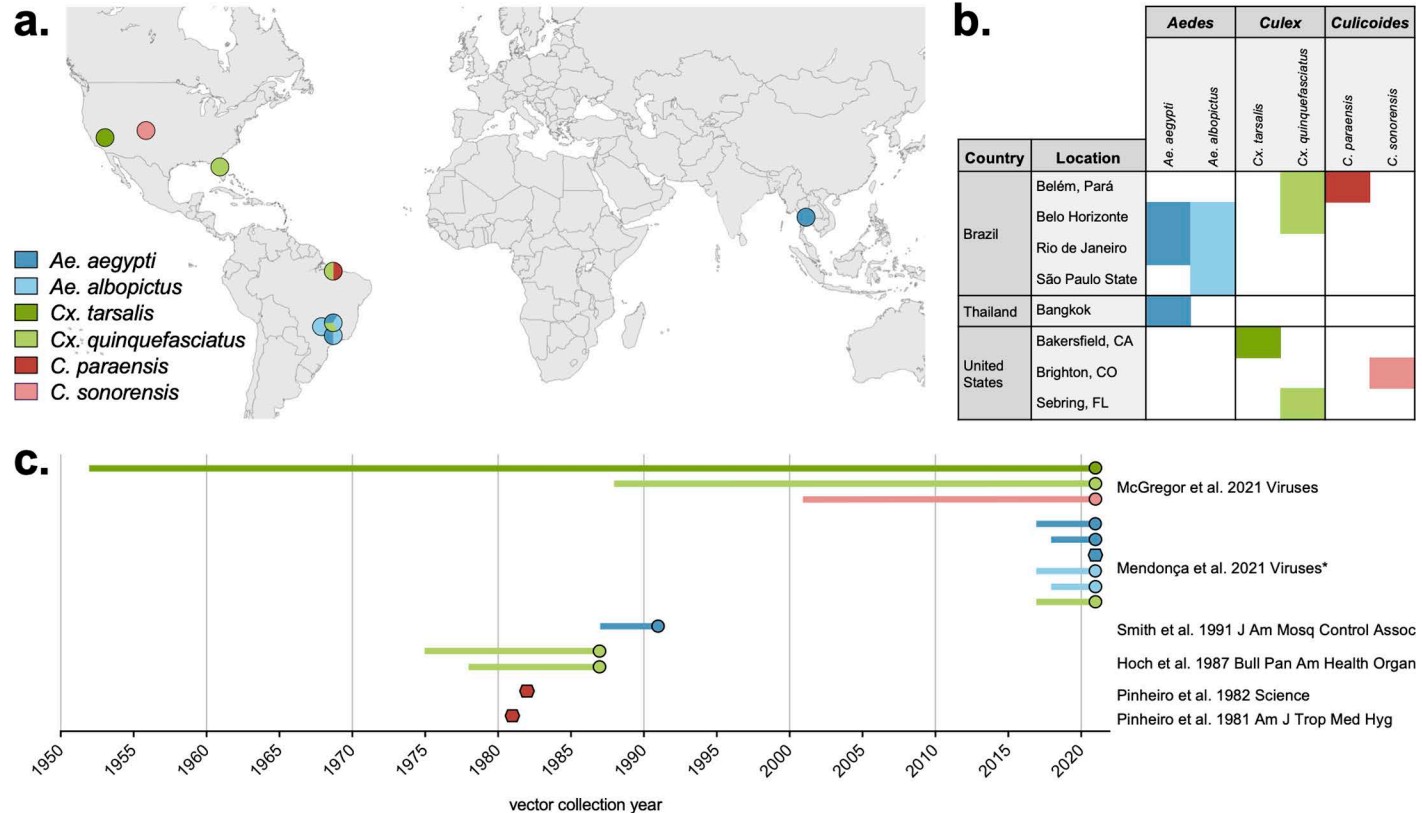

**Fig 3. Experimental vector origin location and collection year.** a) Map and b) table of the origin of each vector used in experimental vector competence publications. World map created in BioRender. c) The vector collection year of each colony/strain of species (line) relative to the corresponding publication year (circle/hexagon). Vectors without a collection year stated in the publication are shown as hexagons without a corresponding line. *Collection year of vectors used in de Mendonça *et al.* [20] were provided directly by the authors.

The first study evaluated OROV infection and transmission in parenterally inoculated *Ae. scapularis*, *Ae. serratus*, *Cx. quinquefasciatus* (referred to as *Cx. fatigans* in the paper), and *Psorophora ferox* (Fig 6) [11]. Despite small group sizes (n = 7, 4, 4, 5, respectively), two weeks after inoculation, all species became infected, with lowest infection rates in *P. ferox* (Fig 6a). To evaluate transmission, two weeks after inoculation, mosquitoes were fed on 2-day old mice. The mice were immediately blended in diluent, which was then intracranially injected into infant mice, and disease was monitored. While the number of recipient mice exposed and tested was not stated, no transmission occurred (Fig 6b).

## Discussion

Our systematic review revealed that, in the six decades between the discovery of Oropouche virus and the 2023–24 epidemic in the Americas, only seven vector competence studies were published. Even accounting for the possibility that more experiments were conducted but never published, our findings reveal that OROV has been subject to chronic neglect – a pattern that is true more broadly of most orthobunyaviruses, particularly compared to well-studied flaviviruses (e.g., yellow fever virus and dengue virus) and alphaviruses (e.g., chikungunya virus and Mayaro virus) [25]. Nevertheless, the seven studies we identified – and the nine arthropod species they examined – provide a useful starting point for establishing the vectors involved in sylvatic and urban OROV transmission, as well as the basic biology of OROV-vector interactions.

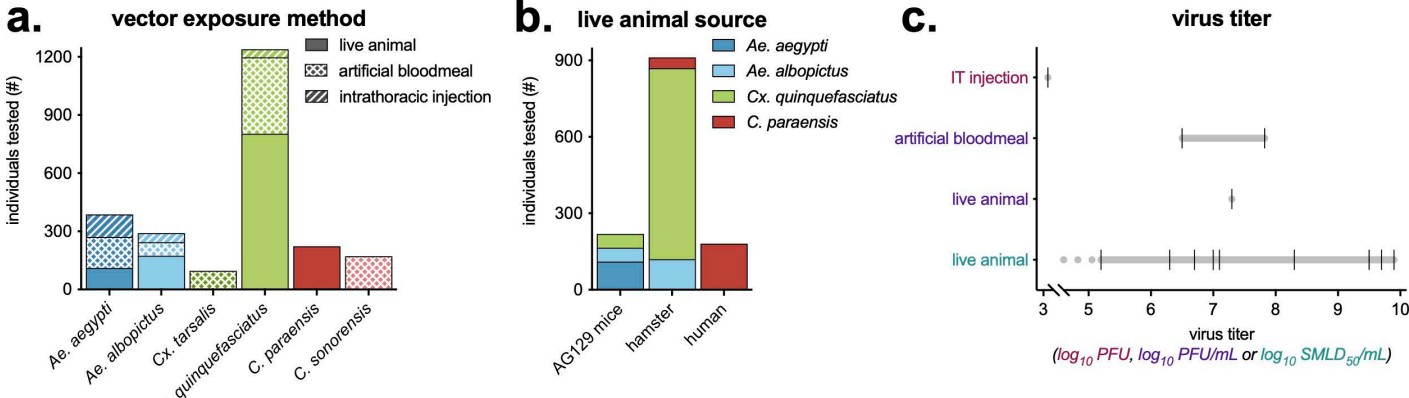

**Fig 4. Vectors and OROV used in vector competence experiments.** a) For each vector species, the total number of individuals tested, and their virus exposure method (e.g., live animal, artificial bloodmeal, intrathoracic injection). b) Of vectors exposed via a live animal, the number of individuals tested by the species of animal they were fed on. c) OROV titer by the vector exposure method and virus quantification assay ($\log_{10}$ transformed plaque forming units (PFU), plaque forming units per milliliter (PFU/mL), or suckling mouse lethal dose 50 per milliliter ($SMLD_{50}$/mL)). OROV titers used in experiments show a black hash mark, with a gray line connecting the range. In one experiment of vectors fed on a live animal, virus was quantified as 5.2 $\log_{10}$ $SMLD_{50}$/mL or lower (denoted as a dotted gray line).

Despite variation in study design, methods, and materials (e.g., geographic origin of vectors or time since collection from the field), results consistently showed that *Aedes* and *Culex* mosquitoes rarely become infected with OROV following an infectious bloodmeal, and have a limited ability to transmit the virus. Intrathoracic injections, which bypass the initial midgut infection and escape barriers and deliver virus directly into the hemocoel, revealed that OROV can replicate in mosquitoes, suggesting lack of infection following oral exposure, is likely not molecular incompatibility between the virus and vector, but instead limited ability to infect the midgut. While studies based on intrathoracic injection are useful to understand fundamental questions of virus-vector interactions, we also caution that they are not representative of natural infection and transmission risk. In nature, vectors will only be exposed to virus orally, and therefore, transmission when these barriers are bypassed (e.g., intrathoracic injection), are not relevant to transmission.

Midges (*Culicoides* spp.) are the primary vectors of many viruses of medical and veterinary importance, including bluetongue virus (BTV), Schmallenberg virus, and OROV-related viruses within the Simbu serogroup of the genus *Orthobunyavirus* [9]. However, they remain dramatically under-studied and under-surveyed in comparison to mosquitoes and ticks. Despite the limited number of publications, all experimental results support the observation that *Culicoides* midges are highly competent vectors for OROV. However, there are few studies experimentally evaluating midge vector competence for any virus due to challenges surrounding lab colonization and experimental manipulation of these arthropods [26]. Epidemiological evidence implicates *C. parensis* as the primary urban vector [7], but detection and isolation rates in wild *C. paraensis* pools have been low (4 positive of 31,555 tested, ~0.01% positivity) [6,7]. Importantly, while there are over 1,300 species of *Culicoides*, OROV has only been detected in *C. paraensis* during outbreak vector surveillance, and only two species (*C. paraensis* and *C. sonorensis*) have been experimentally tested and demonstrated to transmit OROV [7,9]. Additionally, while they are abundant globally, there is far less surveillance of *Culicoides* compared to other vectors (e.g., mosquitoes), leaving incomplete distribution maps, and there are no reported occurrences of any *Culicoides* spp. in Cuba, where OROV transmission is endemic [27]. Further research is needed to confirm which *Culicoides* species are actively involved in the current OROV outbreaks, or could someday pose a risk, especially in new locations.

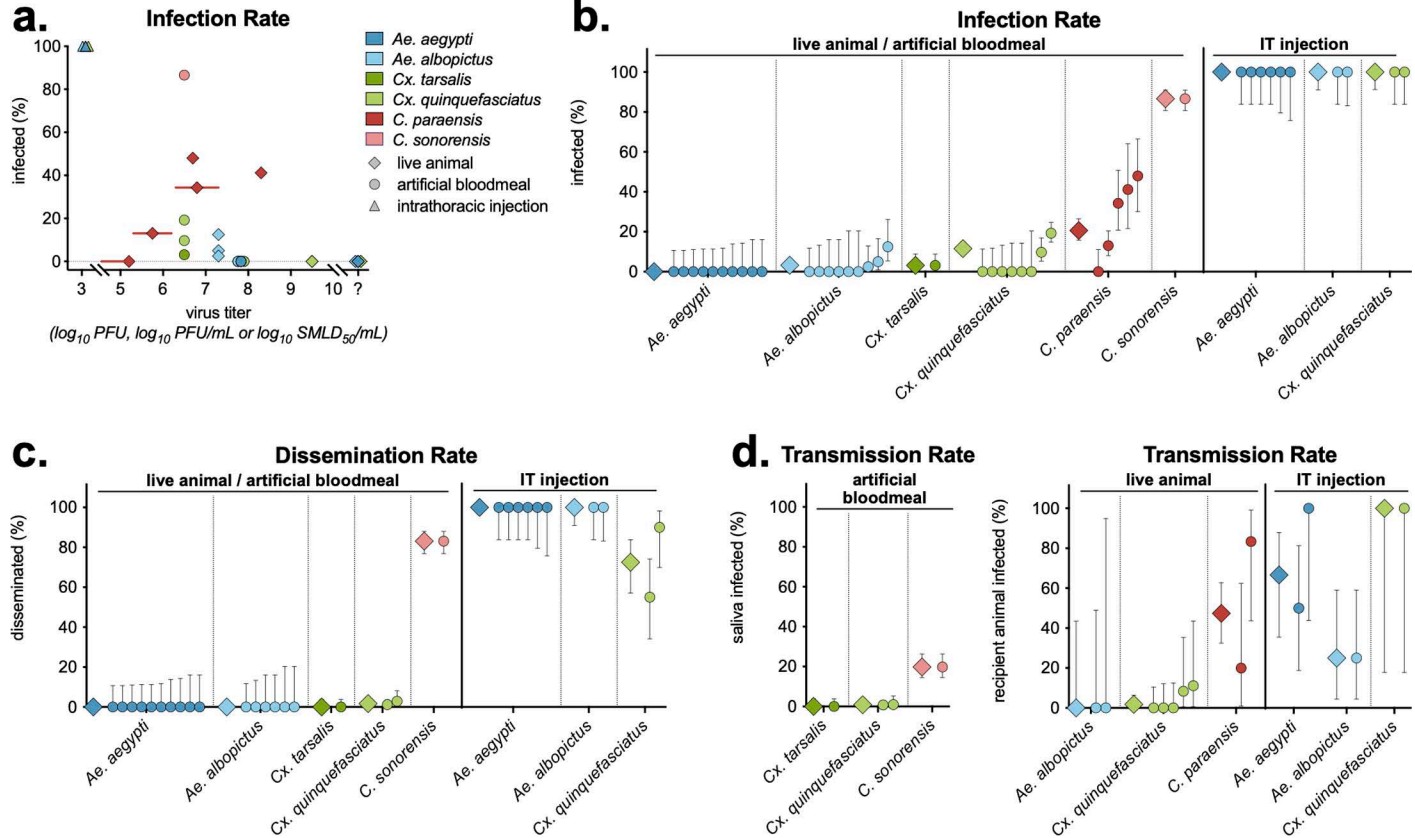

**Fig 5. OROV infection, dissemination and transmission rates in mosquitoes and midges.** Infection, dissemination and transmission rates (percentage of infected vectors of the total tested) for each experiment, by vector spp., a) exposure method and virus titer (all virus quantification assays on the same axis). a) Experiments where virus concentration was unknown are plotted at the far right of the x-axis ("?") (n = 4 *Ae. aegypti*, n = 2 *Ae. albopictus*, n = 2 *Cx. quinquefasciatus*). In three experiments feeding *C. parensis* on live animals, the virus titer was provided as a range (e.g., 5.3-6.3 log$_{10}$ SMLD$_{50}$/mL), shown with a red line with symbol plotted at the average titer. b) Infection, c) dissemination and d) transmission results for all experiments by vector spp. and exposure method. Each experiment is shown as a circle, with averages for the species shown as a diamond. Infection rates (a, b) were measured via vector body or abdomen (artificial bloodmeal and live animal exposure), or head and thorax (intrathoracic injection). c) Dissemination rates were measured via legs, or head and thorax (artificial bloodmeal and live animal), or abdomen (intrathoracic injection). d) Left panel - transmission rates were measured via the number of vector saliva samples out of total saliva samples measured. Right panel - transmission rates were measured via the number of virus positive recipient animals fed on by vectors, out of the total of recipient animals tested. In all graphs, symbols show mean, with error bars showing 95% Wilson/Brown confidence intervals.

Based on current evidence, mosquitoes are unlikely to be a primary vector of OROV in natural settings. Despite extensive efforts, OROV has rarely been isolated from mosquito vectors of other arboviruses, even during outbreaks when it has been estimated >15% of the population is infected [7]. Of > 28,000 *Cx. quinquefasciatus* mosquitoes tested during outbreaks over 14 years, only three have tested positive (~0.01%), including a mosquito engorged with blood caught in a hospital ward at the bedside of a viremic patient 1975 in Pará, Brazil [6,7]. Low infection and transmission rates seen in experimental vector competence studies, and incredibly low rates of virus isolation during outbreaks, suggest that *Cx. quinquefasciatus* are likely a comparatively poor vector for OROV. However, vector competence is only one component of vectorial capacity, which also incorporates factors such as vector density and bloodfeeding behavior [28]; therefore, a vector with low competence may still be important at maintaining and transmitting the virus in nature, as has been demonstrated with *Ae. albopictus* and dengue virus [29].

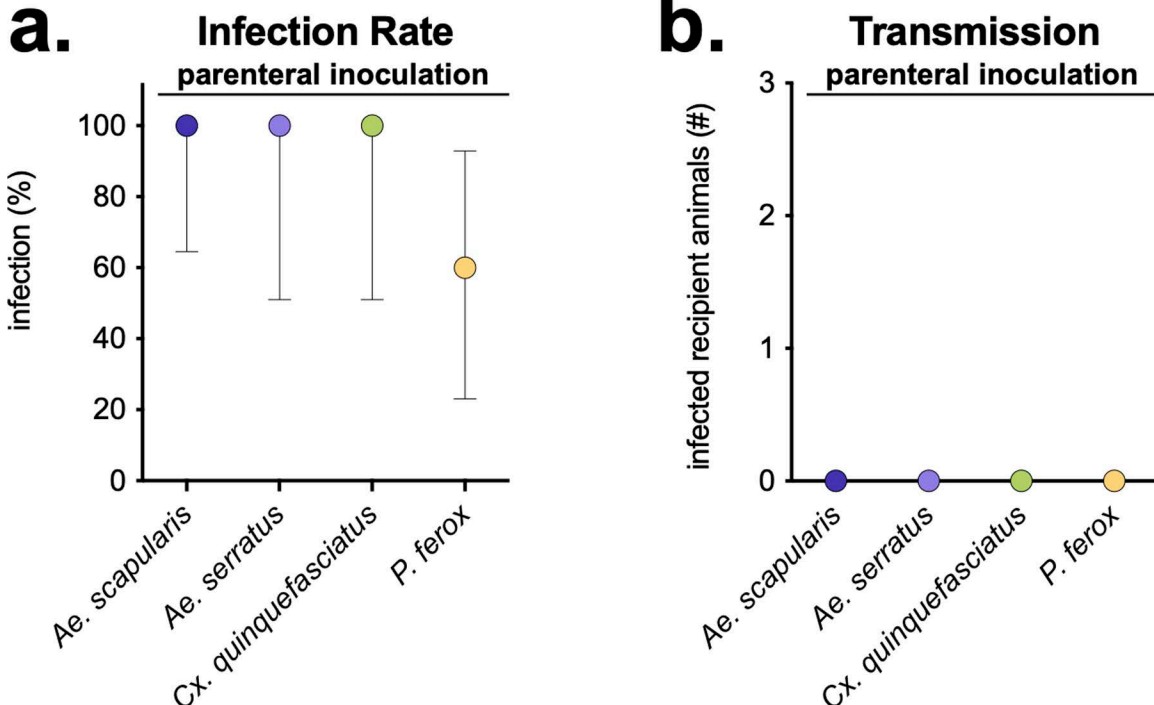

**Fig 6. No OROV transmission in parenterally inoculated mosquitoes.** a) Infection rates (number infected vectors of the total number tested) for each mosquito species. Mosquito samples (e.g., body, thorax) tested to evaluate infection are unknown. Symbols show mean, with error bars showing 95% Wilson/Brown confidence intervals. b) Number of recipient animals testing positive for OROV after i.c. injection of mouse homogenate immediately after being fed on by OROV-inoculated mosquitoes.

It remains unknown whether shifts in vector-virus interactions have contributed to the current epidemic. It is possible that genetic changes to the virus have improved vector competence, similar to what has been seen with chikungunya virus and *Ae. albopictus*, or West Nile virus and *Culex* spp. mosquitoes [30–32]. New experiments should focus on establishing whether the novel reassortant is more transmissible by *Culicoides* midges, or can be newly maintained by common urban vectors such as *Ae. aegypti* or *Cx. quinquefasciatus*. Importantly, a recent paper using OROV isolated from a febrile patient from Cuba in 2024, demonstrated experimental infection rates in mosquitoes (*Ae. albopictus*, *Anopheles quadrimaculatus*, *Cx. quinquefasciatus* and *Cx. pipiens*) were low (<4%), and comparable to rates of mosquitoes infected with the 1955 prototypic strain (<2%), suggesting the currently circulating virus is not adapted to, nor improved at infection of and transmission by mosquitoes [33]. However, other factors could also contribute to the unusual intensity of recent and ongoing outbreaks, including climate change, urbanization, deforestation, and human mobility [10,29,34,35]. Field studies on the epidemiology and drivers of these outbreaks will be an important complement to experimental work, particularly to establish whether Oropouche virus will continue to emerge as a threat to public health in the coming decades.

## Supporting information

**S1 Table. Oropouche virus experimental metadata.**
(XLSX)

**S2 Table. PRISMA checklist.**
(DOCX)

## Acknowledgments

We thank Silvana de Mendonça and colleagues for providing additional data, and thank Doug Brackney and Nate Grubaugh for helpful discussion.

## Author contributions

**Conceptualization:** Emily N Gallichotte, Colin J Carlson.

**Formal analysis:** Emily N Gallichotte.

**Funding acquisition:** Colin J Carlson.

**Investigation:** Emily N Gallichotte.

**Writing – original draft:** Emily N Gallichotte, Colin J Carlson.

**Writing – review & editing:** Emily N Gallichotte, Gregory D Ebel, Colin J Carlson.

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
