## [Decision Letter · Decision Letter 0]

29 Jan 2025

PNTD-D-24-01594Vector competence for Oropouche virus: a systematic review of pre-2024 experimentsPLOS Neglected Tropical Diseases Dear Dr. Gallichotte, Thank you for submitting your manuscript to PLOS Neglected Tropical Diseases. After careful consideration, we feel that it has merit but does not fully meet PLOS Neglected Tropical Diseases's publication criteria as it currently stands. Therefore, we invite you to submit a revised version of the manuscript that addresses the points raised during the review process. Please submit your revised manuscript within 30 days Mar 30 2025 11:59PM. If you will need more time than this to complete your revisions, please reply to this message or contact the journal office at plosntds@plos.org. Please include the following items when submitting your revised manuscript: * A rebuttal letter that responds to each point raised by the editor and reviewer(s). You should upload this letter as a separate file labeled 'Response to Reviewers '. This file does not need to include responses to any formatting updates and technical items listed in the 'Journal Requirements' section below. * A marked-up copy of your manuscript that highlights changes made to the original version. You should upload this as a separate file labeled 'Revised Manuscript with Track Changes '.* An unmarked version of your revised paper without tracked changes. You should upload this as a separate file labeled 'Manuscript '. If you would like to make changes to your financial disclosure, competing interests statement, or data availability statement, please make these updates within the submission form at the time of resubmission. Guidelines for resubmitting your figure files are available below the reviewer comments at the end of this letter. We look forward to receiving your revised manuscript. Kind regards, Olaf Horstick, FFPH(UK)Academic EditorPLOS Neglected Tropical Diseases Andrea MarziSection EditorPLOS Neglected Tropical Diseases

Shaden Kamhawi

co-Editor-in-Chief

Paul Brindley

co-Editor-in-Chief

**Journal Requirements:**

At this stage, the following Authors/Authors require contributions: Emily Gallichotte, Gregory D Ebel, and Colin J Carlson. Please ensure that the full contributions of each author are acknowledged in the "Add/Edit/Remove Authors" section of our submission form.

3) Some material included in your submission may be copyrighted. According to PLOSu2019s copyright policy, authors who use figures or other material (e.g., graphics, clipart, maps) from another author or copyright holder must demonstrate or obtain permission to publish this material under the Creative Commons Attribution 4.0 International (CC BY 4.0) License used by PLOS journals. Please closely review the details of PLOSu2019s copyright requirements here: PLOS Licenses and Copyright. If you need to request permissions from a copyright holder, you may use PLOS's Copyright Content Permission form.

Potential Copyright Issues:

i) Figure 3a. Please (a) provide a direct link to the base layer of the map (i.e., the country or region border shape) and ensure this is also included in the figure legend; and (b) provide a link to the terms of use / license information for the base layer image or shapefile. We cannot publish proprietary or copyrighted maps (e.g. Google Maps, Mapquest) and the terms of use for your map base layer must be compatible with our CC BY 4.0 license.

4) As required by our policy on Data Availability, please ensure your manuscript or supplementary information includes the following:

**Reviewers' comments:** Reviewer's Responses to Questions

**Key Review Criteria Required for Acceptance?**

**Methods**

-Are the objectives of the study clearly articulated with a clear testable hypothesis stated?

-Is the study design appropriate to address the stated objectives?

-Is the population clearly described and appropriate for the hypothesis being tested?

-Is the sample size sufficient to ensure adequate power to address the hypothesis being tested?

-Were correct statistical analysis used to support conclusions?

-Are there concerns about ethical or regulatory requirements being met?

Reviewer #1: Were other search terms considered, such as simply “Oropouche” or “OROV” or other spellings?

Needing raw data available could have been a limitation – was this exclusion criteria applied to any potential publication?

Pg 4: some of the results are repeated here and should be removed (i.e., number of publications found by method). In addition, citation searching was mentioned but not described – were only included publication citations searched?

Pg 4, 98: PubMed

Reviewer #2: Review Methods are appropriate except:

PubMed was the only database used. Methodology for citation searching is not clear.

The term PRISMA is used in figures and supplemental files only.

Reference [1] is a preprint and should not be used as a basis for discussion of changes in clinical presentation.

Comment on recent genetic analyses (Lines 38-40) is based on study by Naveca et al in nature medicine (2025, "Human outbreaks of a novel reassortant Oropouche virus in the Brazilian Amazon region") and needs to be cited.

Reviewer #3: The study methods are well designed and address the specific research objectives: to review vector competence studies for Oropouche virus.

**Results**

-Does the analysis presented match the analysis plan?

-Are the results clearly and completely presented?

-Are the figures (Tables, Images) of sufficient quality for clarity?

Reviewer #1: No concerns

Reviewer #2: Figures mentioning PRISMA without context in the text.

Figure 2a: visualization of publication dates is not clearly labeled

Figure 4: Colors are not defined in legend. Usefulness of this figure is unclear.

Figure 4c: Use of OROV titers is unclear.

Figure 5a: Experiments where virus concentration was unknown are plotted at far right of the x-axis: The legend overlays this portion of the graph and the number of points at "?" are unclear.

Figure 6: This study is so different from the others that its inclusion is questionable. Transmission study was complicated and unclear whether its methodology was appropriate, so viability of the conclusion "No OROV transmission" is in question. Additionally, I needed to combine information from both this figure and lines 140-142 of the text to understand the study methodology utilizing the 2-day old mice. This should have been more clear in the text itself.

Line 51: Reference [7] utilizes more methodology than Culicoides abundance to incriminate midges in transmission.

Lines 47-63: This paragraph needs to contain more data on Culicoides because they have been shown to transmit.

Line 60: conclusion about low OROV prevalence in humans due to low detection rates in mosquitoes is flawed. When these urban vectors were collected in relation to human viremia, and low vector competence more reasonable.

Line 62: Where has speculation about potential vector shift been published?

Line 66: Please briefly describe the components of the data standard and how you used them.

Line 119: "range of OROV quantified" needs better clarification

Lines 140-142: Better communication about this study is needed, as reasoning and sequence of mosquito feeding, baby mouse blending, and IC injection is confusing. It is also worth noting the potential weaknesses of this paper.

Reviewer #3: Results and figures are well done.

**Conclusions**

-Are the conclusions supported by the data presented?

-Are the limitations of analysis clearly described?

-Do the authors discuss how these data can be helpful to advance our understanding of the topic under study?

-Is public health relevance addressed?

Reviewer #1: No concerns

Reviewer #2: Line 158: vector instead of host

Line 158: conclusion that initial infection of midgut cells blocked OROV infection is not supported by data presented in this review. If retained, reference is needed here and presentation of methodology from this reference is needed in the review manuscript.

Lines 165-166: "Culicoides midges are highly competent..." and "few studies evaluating...competence" are conflicting statements. Suggest re-wording for clarity of intention.

Lines 169-170: Review content suggests that Culicoides are already known to actively transmit OROV. "...could someday pose a risk" is not supported and one wonders if this statement was meant for Culex.

Conclusion discusses "shift in vector-virus interactions" (Line 181), which is appropriate. However, Introduction discusses changes in clinical presentation of Oropouche fever (Lines 42-43) which has not been supported. Furthermore, this statement on lines 42-43 is based on a preprint, which should not be used to guide clinical work.

Reviewer #3: Conclusions are clear and summarise the work.

**Editorial and Data Presentation Modifications?**

Reviewer #1: (No Response)

Reviewer #2: Abstract Lines 17-18: "risk of establishment in regions with frequent travel to..." This needs to be re-phrased to allow full thought processes about travel, vector ranges, human infections, animal infections, and spread of pathogens via human travel. At this time, it makes a lot of assumptions.

Summary Line 21: Caution in using the term "major threat" without quantifying current threat or prognosticating future continued threat.

Reviewer #3: Data presentation is excellent, clear and easy to understand. The authors have done well to visually display the different experimental conditions used in each study and the potential impact this would have on the outcomes.

**Summary and General Comments**

Reviewer #1: This manuscript succinctly describes a highly important systematic review to determine the existence of vector competency studies for Oropouche virus. The analysis and interpretation are very good, and the figures are excellent.

Reviewer #2: More text discussion is needed about Culicoides as a vector, since this is currently determined to be the major vector of OROV.

Overall this review is well done and informative. Needs some refining.

Reviewer #3: This is a very well written paper that synthesizes vector competence studies. The authors have done very well summarising the different experimental methods and vector characteristics across studies which can impact the results and comparisons between experimental studies.

As one might expect, the key findings are that we don't know much about which species are potential vectors for Oropouche virus and that much more work is needed in this space. However, because this is a review of all the available studies it would be more impactful to include more information about the vectors that were experimentally competent. Many of the results and discussion were comparing the higher competence in Culicoides to mosquitoes, but aside from an introductory paragraph, I would have liked to know more about about the Culicoides species themselves that did demonstrate competence. It would be good to discuss and compare how those species differ from each other in the experimental studies, and include any other information that may be important for their vectorial capacity such as difference in their distributions, or known seasonality that might be important in the context of Oropouche? Are both of these species known to exist in Cuba, or other places that have historic or current epidemics?

Similarly I was wondering if there is any known evidence on non-human hosts of Oropouche, and if so, it would be good to include a few sentences. It is interesting to note that in L61 you suggest that many blood-feeding arthropods are infected but not all transmit - what could be infecting them with such low transmission in human outside of outbreak years? This is not specific to vector competence, but is important in the context of vectorial capacity, which is in the discussion.

Otherwise this study was excellent and the information is important for designing future studies and interventions for Oropouche virus.

Minor comments:

L14: What constitutes as high? Could you include a value from the results as was done for the mosquitoes

L59: Is this detection rate in humans or mosquitoes - both are mentioned, suggestion to be more explicit.

L141: At first this read to me that the mice were immediately blended (which was not what I expected to see in a vector study this morning), suggestion to be explicit that this is referring to the mosquitoes

L158: remove the brackets at and add another sentence here about why IT injections are not representative of natural infection and transmission risk.

L178: Could you include an example? Ae. albopictus and dengue would be a relevant one.

PLOS authors have the option to publish the peer review history of their article (what does this mean? ). If published, this will include your full peer review and any attached files.

**Do you want your identity to be public for this peer review?** For information about this choice, including consent withdrawal, please see our Privacy Policy .

Reviewer #1: No

Reviewer #2: No

Reviewer #3: No

---

## [Decision Letter · Decision Letter 1]

10 Mar 2025

PNTD-D-24-01594R1Vector competence for Oropouche virus: a systematic review of pre-2024 experimentsPLOS Neglected Tropical Diseases Dear Dr. Gallichotte, Thank you for submitting your manuscript to PLOS Neglected Tropical Diseases. After careful consideration, we feel that it has merit but does not fully meet PLOS Neglected Tropical Diseases's publication criteria as it currently stands. Therefore, we invite you to submit a revised version of the manuscript that addresses the points raised during the review process. Please submit your revised manuscript within 30 days Apr 09 2025 11:59PM. If you will need more time than this to complete your revisions, please reply to this message or contact the journal office at plosntds@plos.org. Please include the following items when submitting your revised manuscript: * A rebuttal letter that responds to each point raised by the editor and reviewer(s). You should upload this letter as a separate file labeled 'Response to Reviewers '. This file does not need to include responses to any formatting updates and technical items listed in the 'Journal Requirements' section below. * A marked-up copy of your manuscript that highlights changes made to the original version. You should upload this as a separate file labeled 'Revised Manuscript with Track Changes '. * An unmarked version of your revised paper without tracked changes. You should upload this as a separate file labeled 'Manuscript '. If you would like to make changes to your financial disclosure, competing interests statement, or data availability statement, please make these updates within the submission form at the time of resubmission. Guidelines for resubmitting your figure files are available below the reviewer comments at the end of this letter. We look forward to receiving your revised manuscript. Kind regards, Olaf Horstick, FFPH(UK)Academic EditorPLOS Neglected Tropical Diseases Andrea MarziSection EditorPLOS Neglected Tropical Diseases

Shaden Kamhawi

co-Editor-in-Chief

Paul Brindley

co-Editor-in-Chief

**Journal Requirements:**

Please ensure that the CRediT author contributions listed for every co-author are completed accurately and in full.

At this stage, the following Authors/Authors require contributions: Emily Gallichotte, Gregory D Ebel, and Colin J Carlson. Please ensure that the full contributions of each author are acknowledged in the "Add/Edit/Remove Authors" section of our submission form.

**Reviewers' comments:** Reviewer's Responses to Questions

**Key Review Criteria Required for Acceptance?**

**Methods:**

-Are the objectives of the study clearly articulated with a clear testable hypothesis stated?

-Is the study design appropriate to address the stated objectives?

-Is the population clearly described and appropriate for the hypothesis being tested?

-Is the sample size sufficient to ensure adequate power to address the hypothesis being tested?

-Were correct statistical analysis used to support conclusions?

-Are there concerns about ethical or regulatory requirements being met?

Reviewer #1: My concerns were mostly addressed, but the actual methods for citation searching are still missing. Were the citations of only included articles searched, or were the citations of non-included articles considered? Was it all articles in that category, or were they targeted somehow?

Reviewer #2: Manuscript is much improved and well done. Great job!

Line22: reads a bit clunky. Since citations are not needed in the Author Summary, recommend simplifying to what you would like to say.

Lines 113-114 dates appear contradictory.

Lines 146-150 is repetitive with line 117 and can be simplified. e.g., "Anderson et al (1961) evaluated OROV infection and transmission in parentally inoculated Ae. scapularis, Ae. serratus, Cx. quinquefasciatus (referred to as Cx. fatigans in the paper) and Psorophora ferox mosquitoes.

Line 154: Please briefly explain what was tested after diluent injection into infant mice

Line 170-171: "block of infection" seems a potential overstatement here. Wording "is not" also comes across strong.

Line 183: Recommend adding the word 'wild', as, "... detection and isolation rates in wild C. paraensis..."

Reviewer #3: (No Response)

**Results:**

-Does the analysis presented match the analysis plan?

-Are the results clearly and completely presented?

-Are the figures (Tables, Images) of sufficient quality for clarity?

Reviewer #1: No more concerns

Reviewer #2: Figure 3: Hexagons appear on the figure between years 1980-1985; but caption states hexagons show data without a collection year

Figure 5: Legend should apply to all graphs in this figure, not just a.

Number of pos samples out of total == Percent of animals tested

Reviewer #3: (No Response)

**Conclusions:**

-Are the conclusions supported by the data presented?

-Are the limitations of analysis clearly described?

-Do the authors discuss how these data can be helpful to advance our understanding of the topic under study?

-Is public health relevance addressed?

Reviewer #1: No concerns

Reviewer #2: Good

Reviewer #3: (No Response)

**Editorial and Data Presentation Modifications?**

Reviewer #1: (No Response)

Reviewer #2: Accept

Reviewer #3: (No Response)

**Summary and General Comments:**

Reviewer #1: (No Response)

Reviewer #2: (No Response)

Reviewer #3: (No Response)

PLOS authors have the option to publish the peer review history of their article (what does this mean? ). If published, this will include your full peer review and any attached files.

**Do you want your identity to be public for this peer review?** For information about this choice, including consent withdrawal, please see our Privacy Policy .

Reviewer #1: No

Reviewer #2: No

Reviewer #3: No

**Figure resubmission:** While revising your submission, please upload your figure files to the Preflight Analysis and Conversion Engine (PACE) digital diagnostic tool, https://pacev2.apexcovantage.com/. PACE helps ensure that figures meet PLOS requirements. To use PACE, you must first register as a user. Registration is free. Then, login and navigate to the UPLOAD tab, where you will find detailed instructions on how to use the tool. If you encounter any issues or have any questions when using PACE, please email PLOS at figures@plos.org. Please note that Supporting Information files do not need this step. If there are other versions of figure files still present in your submission file inventory at resubmission, please replace them with the PACE-processed versions.
---

## [Editor Report · Decision Letter 2]

27 Mar 2025

Dear Dr. Gallichotte,

We are pleased to inform you that your manuscript 'Vector competence for Oropouche virus: a systematic review of pre-2024 experiments' has been provisionally accepted for publication in PLOS Neglected Tropical Diseases.

Best regards,

Olaf Horstick, FFPH(UK)

Academic Editor

Andrea Marzi

Section Editor

Shaden Kamhawi

co-Editor-in-Chief

Paul Brindley

co-Editor-in-Chief

---

## [Editor Report · Acceptance letter]

Dear Dr. Gallichotte,

We are delighted to inform you that your manuscript, "Vector competence for Oropouche virus: a systematic review of pre-2024 experiments," has been formally accepted for publication in PLOS Neglected Tropical Diseases.

Best regards,

Shaden Kamhawi

co-Editor-in-Chief

Paul Brindley

co-Editor-in-Chief
